# Optimization of Scorpion Protein Extraction and Characterization of the Proteins’ Functional Properties

**DOI:** 10.3390/molecules24224103

**Published:** 2019-11-13

**Authors:** Ahmidin Wali, Atikan Wubulikasimu, Sharafitdin Mirzaakhmedov, Yanhua Gao, Adil Omar, Amina Arken, Abulimiti Yili, Haji Akber Aisa

**Affiliations:** 1Key Laboratory of Plant Resources and Chemistry in Arid Regions, Xinjiang Technical, Institute of Physics and Chemistry, Chinese Academy of Sciences, Urumqi 830011, Xinjiang, China; ahmidin@ms.xjb.ac.cn (A.W.); obuatikam@sina.com (A.W.); gaoyh@ms.xjb.ac.cn (Y.G.); adilomar@126.com (A.O.); ak950208@163.com (A.A.); haji@ms.xjb.ac.cn (H.A.A.); 2University of Chinese Academy of Science, Beijing 100039, China; 3Institute of Bioorganic Chemistry, Academy of Sciences of Uzbekistan, Tashkent 100125, Uzbekistan; mirzaakhmedov@mail.ru

**Keywords:** scorpion (*Buthus martensii* Karsch) protein, ultrasonic extraction, response surface methodology, scanning electron microscopy, functional properties

## Abstract

Scorpion has long been used in traditional Chinese medicine, because whole scorpion body extract has anti-cancer, analgesic, anti-thrombotic blood anti-coagulation, immune modulating, anti-epileptic, and other functions. The purpose of this study was to find an efficient extraction method and investigate some of physical and chemical parameters, like water solubility, emulsification, foaming properties, and oil-holding capacity of obtained scorpion proteins. Response surface methodology (RSM) was used for the determination of optimal parameters of ultrasonic extraction (UE). Based on single factor experiments, three factors (ultrasonic power (w), liquid/solid (mL/g) ratio, and extraction time (min)) were used for the determination of scorpion proteins (SPs). The order of the effects of the three factors on the protein content and yield were ultrasonic power > extraction time > liquid/solid ratio, and the optimum conditions of extraction proteins were as follows: extraction time = 50.00 min, ultrasonic power = 400.00 w, and liquid/solid ratio = 18.00 mL/g. For the optimal conditions, the protein content of the ultrasonic extraction and yield were 78.94% and 24.80%, respectively. The solubility, emulsification and foaming properties, and water and oil holding capacity of scorpion proteins were investigated. The results of this study suggest that scorpion proteins can be considered as an important ingredient and raw material for the creation of water-soluble supramolecular complexes for drugs.

## 1. Introduction

Scorpions are Chelicerate arthropods and members of the class Arachnida. There are about 1500 known scorpion species, and only 30 among them produce venom considered potentially dangerous to humans, which might be lethal without medical treatment [1,2]. Scorpion venoms are rich sources of complex mixtures of substances, including toxic peptides, free amino acids, enzymes, nucleotides, lipids, amines, mucoproteins, heterocyclic components, and inorganic salts, which affect the ion channels of both excitable and non-excitable cells [3,4]; some scorpion venoms also contains bioactive compounds, like trimethylamine, nucleosides, and betaine. Therefore, it has anti-thrombosis, anticoagulant, fibrinolysis, analgesic, anti-tumor, anti-epileptic, and several pharmacological effects [5,6,7,8]. A number of studies indicate that scorpion venom is comprised of more than 300 toxins, most of which are less than 10 kDa in molecular mass [9]. Scorpions have been used in traditional medicine in Asia and Africa for thousands of years, and large-scale commercial farming has achieved good social and economic benefits [10,11]. The body parts of scorpions are effective for the treatment of cancer [12,13,14]. In this work, our main attention is focused on investigating whole scorpion body extract, not directed to venomous part.

With the emergence of several novel methods for the study and identification of scorpion body parts and venom components, several bioactive peptides have been proved effective to treat a variety of diseases [15,16,17]. The whole body part of a scorpion is used as medicinal material in traditional Chinese medicine. Although there are previous studies on the extraction, isolation, and activity of scorpion toxin [18,19], few reports are available on the extraction and functional evaluation of scorpion total protein. Scorpion bodies are a main medicinal recourse, but few studies on the bioactive peptides from scorpion proteins have been mentioned [20].

Most of the functional characteristics of proteins play a significant role in the protein’s physical and chemical properties in the processing, storage, preparation, and sale phase at different areas of economy [21,22]. Many factors, such as the protein’s size, pH, temperature, protein’s shape, ionic strength, structure, amino acid sequence, composition, and charge distribution are very important factors of protein functionality. The conditions of the proteins at extraction and each stage of purification and drying are factors that must be taken into consideration [23,24]. Therefore, the aim of this study was to investigate some of physical and chemical parameters, like water solubility, emulsification, foaming properties, and oil-holding capacity of scorpion proteins. In order to reach of this goal, response surface methodology for the optimization of three parameters of ultrasonic extraction (UE)—ultrasonic power, solid/liquid ratio, and extraction time—are also investigated.

## 2. Materials and Methods

### 2.1. Materials and Chemicals

Scorpions (*Buthus martensii* Karsch) were collected (about 1000 pieces) in April in the Turpan region, Xinjiang, China, and after killing, they were put into plastic bottles and kept in a refrigerator. A Pierce BCA Protein Assay Kit was purchased from Thermo Scientific; electrophoresis reagents ware purchased from Biosharp Corporation (Beijing, China). All other chemicals were purchased from local suppliers.

The freeze drier was an FDU-1110 (EYELA Company, Tokyo, Japan), the high-speed refrigerated centrifuge was a CR22N (Hitachi Koki Co., Ltd., Tokyo, Japan), the refrigerated centrifuge was an 5417R (Eppendorf, Germany), and the large-capacity oscillator was an HY-8A (Millipore, Burlington, MA, USA). A Spectra Max M5 enzyme labeling analyzer (Bio-Tec Co., Ltd., USA) was also used.

### 2.2. Total Extraction of Proteins from Scorpion Bodies

Ten grams of dried scorpion were ground and separated through a 40-mesh separator. After that, separated powders (8.2 g) were subjected to de-oiling by petroleum ether (150 mL) in a continuous soxhlet extractor, until the solvent became colorless. The final residue was collected and air-dried. According to the following procedure, scorpion proteins (SPs) were extracted by (1) distilled water, (2) sodium chloride (0.5 M), (3) phosphate buffer (PBS; 50 mM, pH = 8.0), and (4) isoelectric precipitation (adjusted pH to 9.0 with 1.0 N NaOH, and adjusted pH to 4.5 with 1.0 N HCl to precipitate SPs) using an ultrasonic extraction method and a stirring extraction method (as a comparison) for 1 h, respectively. This method was used for the first time by us. Scorpion proteins were extracted with an ultrasonic generator, and the ultrasonic power was 200 w. The extraction steps were repeated three times and dialyzed against distilled water for 48 h at 4 °C (F0136-1 Dialysis Membrane, MWCO 1000Da, United States). The obtained dialysates were lyophilized using a freeze drier (FDU-1110, EYELA Company, Tokyo, Japan) and kept at −20 °C until use.

### 2.3. Methods for the Determination of Protein Extraction Indexes

#### 2.3.1. Determination of Protein Contents (%)

The protein content of lyophilized SPs was determined by the BCA (bicinchoninic acid) method [25]. Bovine serum albumin (BSA) was used as a standard. A total of 1.0 mL of distilled water was used to dissolve 1.0 g of sample, which was clarified through centrifugation at 4500× *g* for 15 min at 4 °C. According to the specification of the BCA protein measuring kit, the absorbance was read by an enzyme labeling instrument (Spectra Max M5 enzyme labeling analyzer) at 562 nm. The average value of three parallel measurements was used to calculate the protein content.

#### 2.3.2. Determination of Yields (%)

The yield of SPs was determined using Equation (1):(1)$$Yield\;\left(\%\right)=\;\frac{m_{Se}}{m_{Sd}}\;\times \;100\%$$
where *m**_Se_* is the weight of scorpion protein extraction (mg), and *m**_Sd_* is the weight of the defatted scorpion (mg).

### 2.4. Single-Factor Experiments

The influence of ultrasonic power (w), extraction time (min) and liquid/solid ratio (mL/g) on protein content and yield of the UE were investigated. Based on the ultrasonic extraction and stirring extraction experiments, scorpion protein was extracted with 0.5 M NaCl buffer by the ultrasonic extraction method. The ultrasonic extraction time (20, 30, 40, 50, and 60 min), liquid/solid ratio (5, 10, 15, 20, and 25 mL/g), and ultrasonic power (50, 100, 200, 300, and 400 w) were tested.

### 2.5. Gel Electrophoresis Analysis

Gel electrophoresis (SDS-PAGE) of the SPs was done using 15% acrylamide gel (Bio-Rad Mini-PROREAN Tetra System) [26]. Gel electrophoresis was carried out at 10 v/cm constant voltage. Coomassie Brilliant Blue R-250 dye was used for staining the protein bands for 1.5 h, which were then de-stained in a decoloring solution for 1.5 h via shaking by large capacity oscillator (HY-8A).

### 2.6. Box–Behnken of RSM and Statistical Analysis

RSM is a statistical tool for solving multivariable problems by using reasonable experimental design method and obtaining certain data through experiments, and by analyzing regression equations to determine the optimal technological parameters [27]. In this study, the RSM and Box–Behnken design was used to optimize the extraction conditions. Based on the single-factor experiment, the complete design was made up of 17 runs, and these were done in triplicate to optimize the levels of the selected variables (extraction time, ultrasonic power, and liquid/solid ratio). For the statistical analysis, the three independent variables were coded as *X*_1_, *X*_2_, and *X*_3_, respectively using Equation (2):(2)$$x_{i}\;=\;\frac{X_{i\;}\;-\;X_{0}}{\Delta X_{i}}$$
where *x_i_* is the coded value of an independent variable, *X_i_* is the real value of an independent variable, *X*_0_ is the real value of an independent variable at the center point, and △*X_i_* is the step change value.

The quadratic equation of the variables is as follows:(3)$$Y\;=\;\beta_{0}\;+\;\sum \beta_{i}x_{i}\;+\;\sum \beta_{ii}x_{i}^{2}\;+\;\sum \beta_{ij}x_{i}x_{j}$$
where *Y* is the predicted response variable; *Β*_0_, *β_i_*, *β_ii_*, and *β_ij_* are the constant regression coefficients of the model; and *x_i_* and *x_j_* (*i* = 1, 3; *j* = 1, 3, *i* ≠ *j*) represent the independent variables.

The accuracy and fitness of the above model were evaluated by the coefficient of determination (*R*^2^) and the *F*-value. Based on the above results, the second order polynomial coefficients were undertaken using Design Expert (Version 8.0.6, USA) software. The model was performed to evaluate the analysis of variance (ANOVA).

### 2.7. Functional Properties

The functional properties of de-oiled scorpion flour (DSF), stirring extraction (SE), and ultrasonic extraction (UE) have been tested at the optimal conditions.

#### 2.7.1. Protein Solubility Analyses

Using the method from [28], with slight modification, the solubility of the protein was determined. A total of 200 mg of SP was dissolved in 20 mL deionized water, and the pH was adjusted to 2, 3, 4, 5, 6, 7, 8, 9, 10, 11, and 12 by 1.0 mol/L HCl and 1.0 mol/L NaOH. After the pH was stabilized, it was stirred for 0.5 h at a speed of 8000 rpm/min for 20 min; centrifugation was done to obtain the supernatant. Using the Bradford method [29], the protein content of the supernatant was determined. The value of each sample was measured three times, and the average value was taken. Protein solubility was calculated using the following formula:(4)$$Protein\;\mathrm{Solubility}\;\%\;=\;\left(\frac{m}{m_{T}}\right)\;\times \;100\%$$
where *m* is the protein content of the supernatant (mg/g), and *m_T_* is the content of total protein in SP (mg/g).

#### 2.7.2. Foaming Properties

The foaming capacity (FC) and foaming stability (FS) were carried out using the methods described previously, with minor modifications [24,30]. Using these methods, 0.2 g of SP was dissolved in 20mL of distilled water (*w*/*v* = 1.0%), and the solution (*V*_0_) was stirred for 2.0 min. The mixture was transferred immediately into a 50 mL tube, the initial foam volume was measured (*V*_1_), and the foam volume after standing for 30 min was also measured (*V*_3_). The FC was calculated using Equation (5): (5)$$\mathrm{FC}\;\left(\%\right)\;=\;\left(\frac{V_{1}\;+\;V_{2}\;-\;V_{0}}{V_{0}}\right)\;\times \;100\%$$
where *V*_2_ is the volume of the liquid remaining just after stirring.

FS is calculated using the following formula:(6)$$\mathrm{FS}\;\left(\%\right)\;=\;\;\frac{V_{3}}{V_{1}}\;\times \;100\%.$$

#### 2.7.3. Water Absorption Capacity (WAC)

Using the method from [31], with slight modifications, water absorption capacity (WAC) was determined. Here, 1 g (*W*_0_) of SP was weighed together with the centrifugal tube (*W*_1_). Thereafter, 10 mL of distilled water was added in the tube and mixed. After being held at room temperature for 30 min, the emulsion was centrifuged at 2100× *g* for 15 min. In the end, the supernatant was decanted, and the tube with sediment was weighed (*W*_2_). Using Equation (7), the WAC was calculated:(7)$$\mathrm{WAC}\;=\;\frac{W_{2}\;-\;W_{1}}{W_{0}}.$$

#### 2.7.4. Oil Absorption Capacity (OAC)

The oil absorption capacity (OAC) was determined using the method reported previously [31], with minor modifications. One gram (*F*_0_) of SP was weighed together with a centrifugal tube (*F*_1_) and mixed with 5 mL of corn oil. The emulsion was incubated at room temperature for 30 min, and then centrifuged (Eppendorf centrifuge 4530R, Germany) at 3500 rpm/min for 15 min under the same conditions. The tube was reweighed (*F*_2_) after the removal of the supernatant. The OAC was determined using Equation (8):(8)$$\mathrm{OAC}\;=\;\frac{F_{2\;}-\;F_{1}}{F_{0}}$$

#### 2.7.5. Emulsifying Properties

The emulsifying activity (EA) of the UE and SE was determined by the method described by Wu et al. [32], with slight modification. Here, 1.0 g of each SP was weighed and dissolved in 20 mL of distilled water and 20 mL of corn oil, and mixed at a high speed for 1 min at room temperature. The mixture was centrifuged at 4000 rpm for 5 min. The EA was calculated using Equation (9): (9)$$\mathrm{EA}\;\left(\%\right)=\;\frac{h}{H}\;\times \;100\%$$
where *h* is the height of the emulsified layer and *H* is the height of the tube contents.

The emulsion stability (ES) of UE and SE were determined using the method from [30], with slight modifications. One gram of each SP was weighed and dissolved with 20 mL of distilled water and 20 mL of corn oil, and thereafter mixed at a high speed for 2 min at room temperature. The mixture was centrifuged at 4000 rpm for 5 min. After preparation, 30 mL of emulsions was then transferred into test tubes, and the emulsions were stored at room temperature; they separated into a top oil layer and a bottom serum layer over time. The ES was evaluated using Equation (10):(10)$$\mathrm{ES}\;\left(\%\right)=\;\frac{H_{s}}{H_{t}}\;\times \;100\%$$
where *H_s_* is the height of bottom serum layer (mm) and *H_t_* is the total height of emulsion in the tube (mm).

### 2.8. Scanning Electron Microscopy (SEM) Analysis

To study the effect of scorpion protein extraction technology on protein content and yield, we analyzed SPs by scanning electron microscopy (SEM; SUPRA 55VP, ZEISS). The morphology and surface characteristics of the SPs were observed and recorded by SEM after the samples were fixed on silicon wafers and sputtered by gold.

### 2.9. Statistical Analysis

The experiments were performed with three independent trials, and all the determinations were triplicated. The results were represented as mean ± standard deviation. Analysis of variance (ANOVA) was performed to identify significant differences (*p* < 0.05).

## 3. Results and Discussion

### 3.1. Analysis of Scorpion Protein Extraction Method

In this study, the effects of diverse buffer solutions for the extraction of scorpion protein were studied. Meanwhile, the effects of ultrasonic extraction (this method is used to increase the yield of protein), as well as stirring extraction of the yield (%) and protein content (%) were compared. Figure 1 and Table 1 show that the order of the effects of four buffer solutions on yield and protein content was 0.5 M NaCl > 20 mM PBS > 0.02 M NaOH > water. The 0.5 M NaCl buffer solution (yield 14.64 ± 0.08%, protein content 79.06 ± 0.05%) with ultrasonic was better than other buffers for extracting scorpion protein, followed by 20 mM PBS (yield 18.29 ± 0.05%, protein content 60.98 ± 0.07%).

### 3.2. Analysis of a Single Factor Results

#### 3.2.1. Effect of Extraction Time on Yield and Protein Content

Extraction time plays a key role in protein extraction [32]. In this experiment, the effect of extraction time on the yield and protein content is examined. For that, ultrasound power was 200 W, the liquid/solid ratio was 15, and extraction time was carried out in a range between 20–60 min. The results obtained are shown in Figure 2A. Obtained data show that with the increase of ultrasonic extraction time, the yield and protein content rapidly increased and reached a maximum at 50 min, at 17.95% and 66.35%, respectively. A further increase of time did not affect the yield and amount of protein.

#### 3.2.2. Effect of Ultrasonic Power on Yield and Protein Content

In this experiment, the effect of ultrasonic power on the yield and protein content was examined. For that, the extraction time was 50 min, the liquid/solid ratio (mL/g) was 15, and the ultrasonic power was evaluated in the range 50–400 w. The results obtained are shown in Figure 2B. The obtained data show that with an increase in power, the yield and protein content increased rapidly, and reached their maximums at 300 w, at 23.60% and 75.30%, respectively; in addition, a further increase of time did not affect the yield and amount of protein.

#### 3.2.3. Effect of Liquid/Solid Ratio on Yield and Protein Content

In this experiment, the effect of the liquid/solid ratio on the yield and protein content is examined. For that, ultrasound power was 200 w, extraction time was 50 min, and the liquid/solid ratio was evaluated in the range between 5–25 mL/g. The results obtained are shown in Figure 2C. The obtained data show that with the increase of liquid/solid ratio to the 15 mL/g the yield, protein content rapidly increased, and when the ratio reached to 20 mL/g, a slower increase can be seen, reaching maximums of 19.21% and 55.32%, respectively. A further increase of time did not affect the yield and amount of protein.

### 3.3. Optimization of Extraction Parameters by RSM

Table 2 shows that there was a significant change in protein content and yield at different values of the selected parameters. Using Equation (3), the results were analyzed. After multiple regressions fitting with Design-Expert V8.0.6 software, the regression model equation is as follows:(11)$$Y_{1}=56.43+7.25A+0.066B+7.69C+4.80AB+4.97AC\;+4.67BC+2.14A^{2}+1.28B^{2}-3.18C^{2}$$
(12)$$Y_{2}\;=\;19.46\;+\;0.86A\;+\;0.75B\;+\;1.93C\;+\;1.14AB\;+\;1.80AC\;+\;1.72BC+\;0.31A^{2}-1.64B^{2}-0.26C^{2}$$
where *Y*_1_ and *Y*_2_ are the predicted conversion (%) of protein content and yield, respectively; and *A*, *B*, and *C* are the extraction time (*X*_1_, min), liquid/solid ratio (*X*_2_, mL/g), and ultrasonic power (*X*_3_, w), respectively.

When the factor sign is positive, it shows that the amount of the response variable increases with an increase in its value, and vice versa. Figure 3A,C can also be used to compare the model predicted and the experimental results. Figure 3B,D is the normal probability chart indicating that the points follow a narrow linear pattern. The analysis of variance of the regression equation is shown in Table 3. *F* is the value of the regression model, where the protein content and yield were 17.49 and 34.12 and the *p*-values were 0.0005 and <0.0001, respectively. These values show that the model obtained was significant. The *F*-value and *p*-value for the lack of fit of the protein content and yield were 1.54 and 2.04, respectively, and 0.3354 and 0.2506, respectively, which shows that the equation has a good fitting degree and high reliability. In addition, the decision coefficients (*R*^2^) of the model were 0.9574 and 0.9777, respectively, and the adjusted coefficients of determination (Adj-*R*^2^) were 0.9027 and 0.9491, respectively. The values of *R*^2^ and adjusted *R*^2^ for the models are shown in Table 3, which shows that the regression equation can predict the result of extracting scorpion protein accurately.

#### 3.3.1. Analysis of the Influence of Various Factors on the Extraction of Scorpion Proteins

The degree of influence of each factor on the test index can be compared by the *F*-value (Table 4): *F* (*X*_1_) = 53.50, 23.98; *F* (*X*_2_) = 4.46, 18.15; *F* (*X*_3_) = 60.17, 118.71. These values indicate that the order of influence of each factor on the extraction of scorpion protein was ultrasonic power (w) > extraction time (min) > liquid/solid ratio (mL/g). The response surface can describe the interaction between variables and predict the optimal conditional values of each factor [33]. Figure 4A–F shows the effect of operational variables on protein content and yield protein content and yield. The effect of extraction time on the protein process is greater than that of the ratio of the liquid/solid, and that of ultrasonic power is greater than that of extraction time. According to the above analysis, the order of the influence of each variable on the extraction technology of scorpion protein was ultrasonic power > extraction time > liquid/solid ratio.

#### 3.3.2. Interactions of Variables

Contour plots for each of the fitted models that display the effects of the three variables (to visualize the combined effects of the three factors on protein content and yield) were generated. Figure 4 illustrates the two-dimensional (2D) plots of the binary interactions of the variables on protein content and yield via the contour plots. The interaction between the extraction time and the liquid/solid ratio is shown in Figure 4A,D. This plot indicates that protein content and yield depend more on *X*_1_ than on *X*_2_. Figure 4A,D reveals that at low values of *X*_1_, maximum protein content and yield occurs at higher values of *X*_2_. However, at higher values of *X*_1_, maximum protein content and yield occurs at lower values of *X*_2_. As seen in Figure 4A,D, the interaction between the two factors is weak. The interaction between the extraction time and ultrasonic power is shown in Figure 4B,E. This plot indicates that protein content and yield depend more on *X*_3_ than on *X*_1_. In Figure 4C,F, the effect of the interaction between the liquid/solid ratio and ultrasonic power is depicted. Figure 4C,F shows that increasing the liquid/solid ratio at different ultrasonic power levels has no important effect on the protein content and yield, so the plot shows that protein content and yield depends more on *X*_3_ than on *X*_2_. Therefore, the optimum values of *X*_1_, *X*_2_, and *X*_3_, as determined by the software, are 50 min, 400 w, and 18 mL/g, respectively.

#### 3.3.3. Determination and Validation of Optimal Extraction Conditions

Optimum conditions, which were obtained by Design-Expert V8.0.6 software, were as follows: extraction time = 47.68 min; ultrasonic power = 395.84 w; and solid/liquid ratios = 18.01 mL/g. In view of the feasibility of the experiment, the optimum conditions of extraction proteins were adjusted as follows: extraction time = 50.00 min, ultrasonic power = 400 w, and solid/liquid ratio = 18.00 mL/g. After several tests (*n* > 3), the protein content and yield were 78.94%, and 24.80%, respectively. It is shown that the regression equation and the optimal conditions obtained by the response surface method are reliable.

### 3.4. Functional Properties of Scorpion Proteins

#### 3.4.1. Protein Solubility (PS)

The protein solubility (PS) profiles of de-oiled scorpion flour (DSF), ultrasonic extraction (UE), and stirring extraction (SE) in the water at different pH ranges (2–12) are shown in Figure 5. The PS of DSF, UE, and SE were significantly different, and Figure 5 shows the same U-shaped curves. When the pH value is in the range of 2–4, the solubility of DSF, UE, and SE decreased; however, when in the pH range of 6–10, the solubility of DSF, UE, and SE significantly increased. The minimum protein solubility of DSF, UE, and SE were presented at pH 4, with values of 8.05%, 15.25%, and 18.75%, respectively. The maximum protein solubility was presented at pH 12, with values of 13.5%, 70.15%, and 79.5%, respectively. Therefore, results indicate that scorpion protein extracted by ultrasonic method shows exceptional solubility at an alkaline pH.

#### 3.4.2. Water and Oil Absorption Capacity

The obtained results for the water absorption capacity (WAC) and oil absorption capacity (OAC) of proteins extracted by ultrasonic and stirring are presented in Table 4. The WAC and OAC from both methods were significantly different (*p* < 0.05), with UE having the higher WAC (40.3 ± 0.28 g/g) and OAC (27.70 ± 0.14 g/g) than SE (with WAC = 33.45 ± 0.07 g/g and OAC = 18.80 ± 0.15 g/g) at pH 7.0.

#### 3.4.3. Emulsifying Properties

Charged or non-charged polar amino acids in proteins can affect the proteins’ emulsifying and surfactant properties. Results of emulsifying properties of UE and SE are shown in Table 4. At pH 7.0, the emulsifying properties of UE and SE were significantly different from one another, with values of 45.55 ± 0.64% and 40.25 ± 0.07%, respectively. However, the emulsifying property of UE was significantly different and was higher (85.50 ± 0.28%) than that of SE (69.45 ± 0.35%). The UE had pronounced effects on the emulsifying properties, since it might be exposing more hydrophobic groups to the water and oil interface, giving rise to increased emulsifying capacity and more stable emulsion.

#### 3.4.4. Foaming Properties

Foam formation is similar to emulsion formation. In foam formation, water molecules surround air droplets, which is a nonpolar phase. At pH 7, the foam capacity (FC) of UE was considerably higher than that of SE, with values of 40.30 ± 0.28% and 33.45 ± 0.07%, respectively. The foam stability (FS) of UE was significantly higher than that of SE, with values of 21.70 ± 0.14% and 18.80 ± 0.15%, respectively (Table 4). The results obtained show that the UE, compared to SE, has more flexible protein structure in aqueous solution, and forms more stable foams for stronger interaction on the air–water interface. The variations in foaming properties of UE and SE might be explained with the difference in their protein concentrations. A high protein concentration will improve foam capacity and stability, increasing the viscosity and promoting the formation of the interfacial multilayer membrane [34].

### 3.5. Scanning Electron Microscopy Analysis

Protein samples obtained using various extraction methods were investigated by the SEM method. Figure 6 shows the surface states of protein samples extracted from whole-body, de-oiled protein (DSF) powder, stirred extraction (SE), and ultrasonic extraction (UE). These studies focused on the study of the surface layer of protein molecules. Accordingly, in Figure 6A, adhesive surface layers of protein molecules in dried DSF can be seen. This state has a collapsed form, which shows the density of protein molecules in relation to one other; therefore, these molecules have a higher crystalline degree than other states, and this will result in lower water solubility. In Figure 6B, on the surface of dried SE protein layers, a plurality of elements similar to “wood shavings” can be seen, in contrast to Figure 6A. These elements formed as a result of separation from the collapsed surface. This means that during drying after SE, from the crystalline state shown in Figure 6A, the protein molecules transition from a crystalline to amorphous state, as shown in Figure 6B; the outcome of this results in an increase in the solubility of the protein molecules in water. Also, in Figure 6C, you can see the results of the drying of protein molecules obtained after UE. This picture clearly shows the increase the number of “wood shavings” on the surface, compared to Figure 6B, and how the transformation collapse forms into thin film pieces. Therefore, UE leads to the formation of scorpion protein molecules from a crystalline state to an amorphous one, which results in an increase of their water solubility.

## 4. Conclusions

Studies have shown that the sum of proteins extracted from scorpion bodies had good emulsifying and foaming properties. At the same time, when they were subjected to extraction under different conditions, UE was found to be the best way to obtain the sum of proteins with improved water solubility. According to SEM analysis, the improvement of water solubility of the sum of protein components is accompanied by the transition of these macromolecules from the collapsed and crystalline form to a more amorphous and friable phase. For the extraction of scorpion body parts, some simple ways, i.e., cold water, hot water, and ethanol extraction methods used previously [35,36] were used, and SPs obtained by these methods were physiologically effective for the treatment of various diseases. However, we think that using hot water or ethanol solvents may generally result in the denaturation of native compounds and reduce the real activities of the extracted SPs. Therefore, using UE in an alternative way, at the optimal conditions of UE (i.e., *X*_1_, *X*_2_, and *X*_3_ are 50 min, 400 w, and 18 mL/g, respectively) resulted in increased water solubility and improvement of its emulsifying and foaming properties. In our opinion, the SPs obtained by the UE way should keep a number of their above-mentioned activities, and will be planned for use in creating medicines on that basis in TCM. On the other hand, UE could be the basis for the creation of water-soluble supramolecular complexes for those drugs that are poorly soluble in water. Obtaining such drugs in this way will contribute in the future to (1) an increase the drugs’ water solubility, (2) a reduction their therapeutic doses, (c) increase pharmacological action, and (d) reduce toxicity and side effects. Since the sum of proteins from the scorpion body are not individual components, it might be promising, if created in their base gel drug forms, that they could be used through the skin.

## Figures and Tables

**Figure 1 molecules-24-04103-f001:**
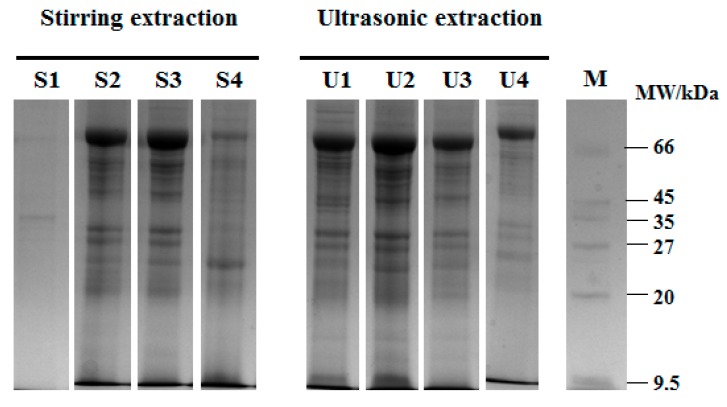
SDS-PAGE electrophoresis of scorpion proteins with different extractions and buffer conditions. M refers to the marker; S1, S2, S3, and S4 are scorpion proteins extracted by stirring with water, 0.5 M NaCl, 20 mM phosphate buffer (PBS), alkali extraction, and isoelectric precipitation, respectively; U1, U2, U3, and U4 are scorpion proteins extracted by ultrasonic with water, 0.5 M NaCl, 20 mM PBS, alkali extraction, and isoelectric precipitation, respectively.

**Figure 2 molecules-24-04103-f002:**
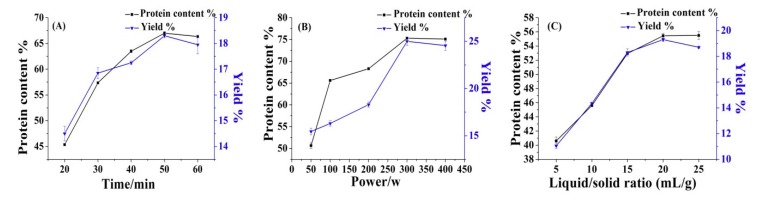
Effects of extraction time (**A**), ultrasonic power (**B**), and liquid/solid ratio (**C**) on yield and protein content.

**Figure 3 molecules-24-04103-f003:**
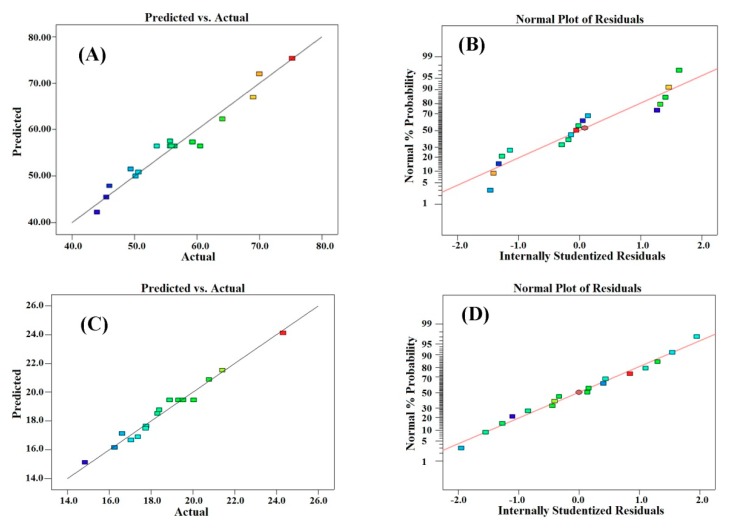
Standard statistical diagrams for model verification. (**A**,**C**) Model-predicted values versus actual data. This figure shows a comparison between the model-predicted values and the actual values that have been obtained by the experiments. (**B**,**D**) Normal probability plot of the residuals. Monotonous distribution and linearity of the data, which are obvious from these figures, confirm the validity of the model and its capability to predict the results.

**Figure 4 molecules-24-04103-f004:**
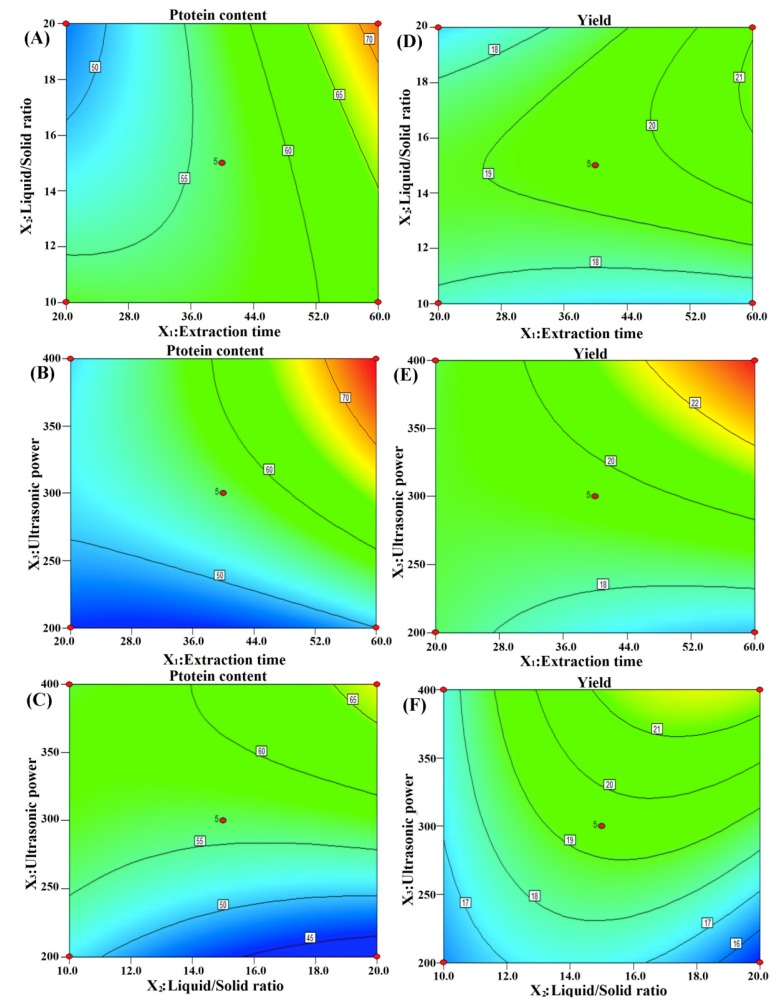
Effects of operational variables on protein content and yield. Interaction between (**A**) liquid/solid ratio and extraction time, (**B**) ultrasonic power and extraction time, and (**C**) ultrasonic power and liquid/solid ratio to protein content. Also depicted are the interaction between the (**D**) liquid/solid ratio and extraction time, (**E**) ultrasonic power and extraction time, and (**F**) ultrasonic power and liquid/solid ratio to yield.

**Figure 5 molecules-24-04103-f005:**
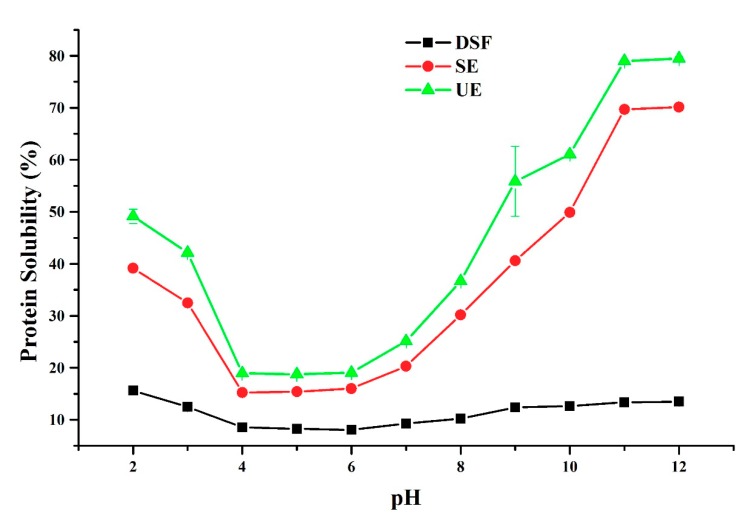
Protein solubility profiles of de-oiled scorpion flour (DSF), stirring extraction (SE), and ultrasonic extraction (UE) at different pH levels.

**Figure 6 molecules-24-04103-f006:**
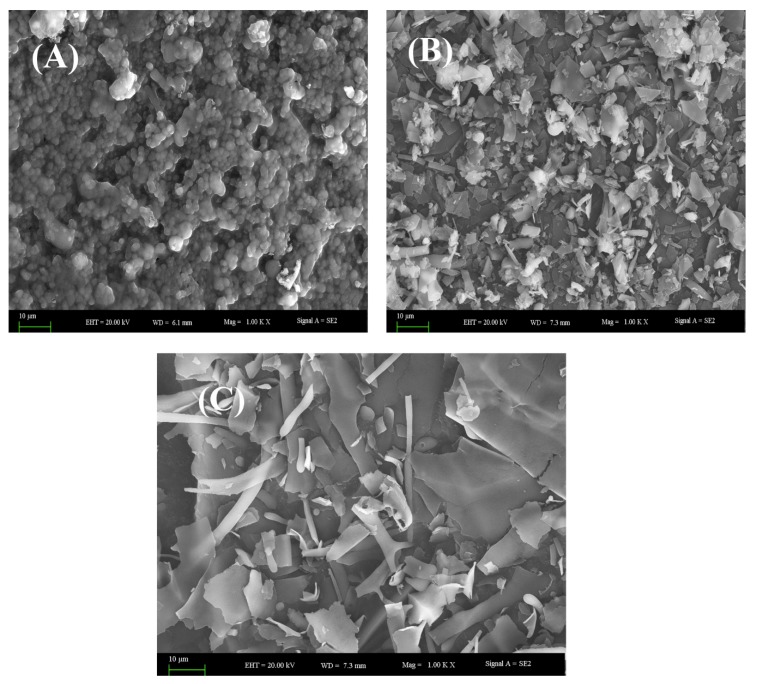
Scanning electron microscope images of (**A**) DSF, (**B**) SE, and (**C**) UE. (1.00 KX magnifications, bar 10 μm).

**Table 1 molecules-24-04103-t001:** Effects of ultrasonic and stirring methods on extraction of total proteins from scorpion body.

Extraction Method	Yield (%)		Protein Content (%)	
	Ultrasonic Extraction	Stirring Extraction	Ultrasonic Extraction	Stirring Extraction
Water	34.85 ± 0.06	34.15 ± 0.09	31.14 ± 0.04	18.08 ± 0.06
0.5 M NaCl	14.64 ± 0.08	11.80 ± 0.03	79.06 ± 0.05	35.26 ± 0.08
20 mM PBS	18.29 ± 0.05	13.45 ± 0.10	60.98 ± 0.07	37.25 ± 0.09
0.02 M NaOH	7.70 ± 0.11	7.57 ± 0.13	50.87 ± 0.07	51.91± 0.17

**Table 2 molecules-24-04103-t002:** RSM experimental design and results for the three-factor/three-level Box–Behnken design (BBD).

Runs	Extraction Time (min) *X*_1_	Liquid/Solid Ratio (mL/g) *X*_2_	Ultrasonic Power (W) *X*_3_	Protein Content/%	Yield /%
1	60	15	400	75.23	24.32
2	40	20	400	69.01	21.41
3	20	10	300	59.31	17.76
4	40	15	300	55.98	19.53
5	40	15	300	56.39	18.89
6	60	10	300	64.09	16.62
7	40	20	200	44.01	14.84
8	40	10	200	49.4	16.25
9	60	15	200	50.17	17.04
10	40	10	400	55.70	17.74
11	40	15	300	55.70	19.31
12	60	20	300	70.00	20.78
13	40	15	300	53.58	19.52
14	20	20	300	46.01	17.37
15	20	15	200	45.49	18.31
16	20	15	400	50.66	18.39
17	40	15	300	60.52	20.04

**Table 3 molecules-24-04103-t003:** Least-squares fit and parameter estimates (significance of regression coefficient).

Source	Protein Content		Yield	
	*F*-Value	*p*-Value	*F*-Value	*p*-Value
Model	17.49	0.0005	34.12	< 0.0001
*X* _1_	53.50	0.0002	23.98	0.0018
*X* _2_	4.46	0.9486	18.15	0.0037
*X* _3_	60.17	0.0001	118.71	< 0.0001
*X* _1_ _× 2_	11.73	0.0111	20.67	0.0026
*X* _1_ _× 3_	12.57	0.0094	51.76	0.0002
*X* _2_ _× 3_	11.11	0.0125	25.77	0.0014
*X* _1_ ^2^	2.45	0.1617	1.67	0.2378
*X* _2_ ^2^	0.88	0.3800	45.25	0.0003
*X* _3_ ^2^	5.43	0.0526	1.12	0.3256
Lack of Fit	1.54	0.3354	2.04	0.2506
*R* ^2^	0.9574		0.9777	
Adj-*R*^2^	0.9027		0.9491	

**Table 4 molecules-24-04103-t004:** Functional properties of scorpion proteins.

Properties	Ultrasonic Extraction	Stirring Extraction
Water holding capacity (g/g)	25.25 ± 0.21	20.45 ± 0.07
Oil holding capacity (g/g)	38.50 ± 0.14	30.65 ± 0.64
Emulsifying activity (%)	45.55 ± 0.64	40.25 ± 0.07
Emulsion stability (%)	85.50 ± 0.28	69.45 ± 0.35
Foam capacity (%)	40.30 ± 0.28	33.45 ± 0.07
Foam stability (%)	21.70 ± 0.14	18.80 ± 0.15

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
