# Peer review of "Optimization of Scorpion Protein Extraction and Characterization of the Proteins’ Functional Properties"

_molecules, 2019, doi:10.3390/molecules24224103_

Round 1

Reviewer 1 Report

Major revision

Although the manuscript is well planned and organized, some major points have to be considered in order to be accepted in Molecules.

Lines 33-35: These sentences would be rewritten in order to avoid repetitions “Scorpion venom contains bioactive compounds, such as trimethylamine, nucleosides and betaine. Scorpion venom has pharmacological effects of anti-thrombosis, anticoagulant, fibrinolysis, analgesic, anti-tumor, and anti-epileptic [3-6]. Scorpion venoms are rich source of complex mixtures”

Lines 46-47: Please, provide references “Although there are previous studies on the extraction, isolation and activity of scorpion toxin”

Lines 48-49: Which bioactive compounds? Please add references “Scorpion bodies are main medicine recourse, but little researches on the bioactive components from scorpion protein have been mentioned.”

Lines 56-58: The authors should include the aim of this purpose.

Line 60: Materials and chemicals: The authors should add more information in this section.

Line 64: 2.2. Extracting proteins from scorpion body: This method is based on literature or the authors have optimised it?

Line 79: Please, provide x g instead of rpm

Line 84: Please uniform the style of parameters: in equation is mSe and mSd and in the explanation se and sd.

Some references are in blue and others in black (and also equations’ numbers and figures)

Line 172: The authors should include the reason of using ultrasound (better yield, etc.)

Line 181: Figure 1 caption. I suggest “SDS-PAGE electrophoresis of scorpion protein with different extractions and buffer conditions“ instead of 15% SDS-PAGE electrophoresis of scorpion protein was extracted with different buffer solution.

Line 186: Please, improve the title of the table 1 and the tabulation of content.

Line 192: where is table 2 in the text?

Line 244: Improve table 3.

Line 288: Could I understand that the functional properties have been tested on the optimal conditions of extractions? Please, this must be clarified.

Conclusions section are poorly described. The authors should include conclusions about results not about methodology. In this work, it is very important since a lot of results are included.

Author Response

Response to Reviewer 1:

Dear professor:

Thank you for your kind comment on our manuscript. We carefully read your reports and revised the manuscript according to the suggestions. The responses as follows:

Comment 1: Lines 33-35: These sentences would be rewritten in order to avoid repetitions “Scorpion venom contains bioactive compounds, such as trimethylamine, nucleosides and betaine. Scorpion venom has pharmacological effects of anti-thrombosis, anticoagulant, fibrinolysis, analgesic, anti-tumor, and anti-epileptic [3-6]. Scorpion venoms are rich source of complex mixtures”

Edited version: Scorpion venoms are rich source of complex mixtures of substances including toxic peptides, free amino acids, enzymes, nucleotides, lipids, amines, mucoproteins, heterocyclic components, inorganic salts, which affects the ion channels of both excitable and non-excitable cells [7, 8], also has some contains bioactive compounds, like: trimethylamine, nucleosides and betaine. Therefore it has anti-thrombosis, anticoagulant, fibrinolysis, analgesic, anti-tumor, anti-epileptic and several pharmacological effects [3-6].

Comment 2: Lines 46-47: Please, provide references “Although there are previous studies on the extraction, isolation and activity of scorpion toxin”

Edited version: Although there are previous studies on the extraction, isolation and activity of scorpion toxin [18, 19], there are few reports on the extraction and functional evaluation of scorpion total protein. Scorpion bodies are main medicine recourse, but few researches on the bioactive peptides from scorpion protein have been mentioned [20]. We list references 18, 19 and 20.

Comment 3: Lines 48-49: Which bioactive compounds? Please add references “Scorpion bodies are main medicine recourse, but little researches on the bioactive components from scorpion protein have been mentioned.”

Edited version: Scorpion bodies are main medicine recourse, but few researches on the bioactive peptides from scorpion protein have been mentioned [20].

Comment 4: Lines 56-58: The authors should include the aim of this purpose.

Edited version: The conditions of the proteins at extraction and each stage of purification and drying are the factors that must be taken into consideration [23, 24]. Therefore the aim of this study was to investigate some of physical and chemical parameters, like: water soluble, emulsifying, foaming properties and oil-holding capacity of scorpion proteins. In order to reach of this goal, response surface methodology for the optimization of three parameters of UE, ultrasonic power, solid/liquid ratio, and extraction time also investigated.

Comment 5: Line 60: Materials and chemicals: The authors should add more information in this section.

Edited version: Scorpions (Buthus martensii Karsch) were collected (about 1000 pieces) by ourselves in April month in Turpan region, Xinjiang, China and after killing they were put into plastic bottle and kept in refrigerator until use. Pierce BCA Protein Assay Kit was purchased from Thermo Scientific; Electrophoresis reagents ware purchased from Biosharp Corporation (Beijing, china); All other chemicals were purchased from local suppliers.

Freeze drier, FDU-1110, EYELA Company, Japan; High-speed refrigerated centrifuge CR22N Hitachi Koki Co., Ltd. Tokyo Japan; Refrigerated centrifuge 5417R Eppendorf; Large capacity oscillator HY-8A; Millipore, USA; Spectra Max M5 enzyme labeling analyzer, Bio-Tec Co., Ltd, USA.

Comment 6: Line 64: 2.2. Extracting proteins from scorpion body: This method is based on literature or the authors have optimized it?

Response 6: We have optimized it by ourselves.

Comment 7: Line 79: Please, provide x g instead of rpm

Response 7: We have changed (rpm) to (g), which was equal to 4500 g

Comment 8: Line 84: Please uniform the style of parameters: in equation is mSe and mSd and in the explanation se and sd.

Response 8: We have edited it in text.( mSe - weight of scorpion protein extraction; mSd - weight of defatted scorpion)

Comment 9: Some references are in blue and others in black (and also equations’ numbers and figures)

Response 9: All references, figures and equation numbers were marked with black now.

Comment 10: Line 172: The authors should include the reason of using ultrasound (better yield, etc.)

Response 10: It has been rewritten.

Comment 11: Line 181: Figure 1 caption. I suggest “SDS-PAGE electrophoresis of scorpion protein with different extractions and buffer conditions “instead of 15% SDS-PAGE electrophoresis of scorpion protein was extracted with different buffer solution.

Response 11: We have revised it according to your suggestion.

Comment 12: Line 186: Please, improve the title of the table 1 and the tabulation of content.

Response 12: We have improved it. (Table 1 Effects of ultrasonic and stirring methods to extraction of total protein from Scorpion body).

Comment 13: Line 192: where is table 2 in the text?

Response 13: It was additional information only. So, we decided to remove it from manuscript.

Comment 14: Line 244: Improve table 3.

Response 14: We have improved it.

Comment 15: Line 288: Could I understand that the functional properties have been tested on the optimal conditions of extractions? Please, this must be clarified.

Response 15: The functional properties of de-oiled scorpion flour (DSF), stirring extraction (SE) and ultrasonic extraction (UE) have been tested at the optimal conditions. The results of functional properties of UE which were discussed, measured under the optimal conditions.

Comment 16: Conclusions section are poorly described. The authors should include conclusions about results not about methodology. In this work, it is very important since a lot of results are included.

Response 16: The conclusions section is fully changed.

Reviewer 2 Report

The manuscript by Wali et al. describes the extraction of protein from dried scorpion tissue by ultrasonic treatment. While I believe the technical aspects of this paper are fine I have some concerns regarding the justification behind this technical development. Specifically, the authors make claims in the abstract, introduction and discussion (e.g. lines 12-13; lines 39-42; lines 355-357) regarding the medical importance of consuming scorpion proteins, but they provide very little explanation to this claim. They provide some references, but still they should provide more details regarding the basis for their claims. I do not mean any disrespect for traditional medicine, but this is a sensitive topic and scientific work should strive to make the most accurate and fact-based claims. Thus, I think the authors should significantly expand the description of the supporting evidence for their claims or alternatively tone them down. Moreover, to the best of my knowledge much of the usage of scorpion in traditional medicine is specific to venom, but the authors in the current work do not focus on venom at all, but extract proteins from whole scorpion tissue. Please address my concerns and revise the manuscript accordingly.

Author Response

Response to Reviewer 2:

Dear professor:

Thank you for your kind comment on our manuscript. We carefully read your reports and revised the manuscript according to the suggestions.

Also we did some changes to the title, authors and abstract.

Many tanks again for your support

Your commitments:

The manuscript by Wali et al. describes the extraction of protein from dried scorpion tissue by ultrasonic treatment. While I believe the technical aspects of this paper are fine I have some concerns regarding the justification behind this technical development. Specifically, the authors make claims in the abstract, introduction and discussion (e.g. lines 12-13; lines 39-42; lines 355-357) regarding the medical importance of consuming scorpion proteins, but they provide very little explanation to this claim. They provide some references, but still they should provide more details regarding the basis for their claims. I do not mean any disrespect for traditional medicine, but this is a sensitive topic and scientific work should strive to make the most accurate and fact-based claims. Thus, I think the authors should significantly expand the description of the supporting evidence for their claims or alternatively tone them down. Moreover, to the best of my knowledge much of the usage of scorpion in traditional medicine is specific to venom, but the authors in the current work do not focus on venom at all, but extract proteins from whole scorpion tissue. Please address my concerns and revise the manuscript accordingly.

The responses as follows:

Thank you very much for your kind comments and suggestions.

Responding to your commitments we are agree with you and we fully changed context of manuscript, paying more attention to use in medicine. So we have added aim of work, why we are doing this work? and at which are might be useful obtained results? We explained in conclusions. Since most of your commitments were similar to the commitments of reviewer 1, we hope that, we have answered to your suggestions. If you read this manuscript again from the starting point, we hope that you will find responds to your commitments.

Also we did some changes to the title, authors and abstract.

Many tanks again for your support

Reviewer 3 Report

The manuscript entitled "Optimization of ultrasonic extraction of proteins from scorpion and characterization of functional properties" deals with the isolation under optimal conditions of proteins obtained from scorpion bodies. The subject is sound and in general terms the manuscript is well structured and most results are relevant and deserve publication in the Molecules journal.

Nevertheless somo minor points should be considered to improve this manuscript. These are listed below.

Grammar and spelling should be revised all through the text. Dried scorpions were grinded (line 65). Which was the granulometry of samples before extraction? Line 103. Experiments were run in duplicate, why not in triplicate as it is usual in experimental design and statistical analysis? Please clarify In general terms there is some lack of discussion in the correspondinig section. I would encourage authors to compare their results with those obtained by other authors.

Author Response

Response to Reviewer 3:

Dear professor:

Thank you for your kind comment on our manuscript. We carefully read your reports and revised the manuscript according to the suggestions.

Your commitments:

The manuscript entitled "Optimization of ultrasonic extraction of proteins from scorpion and characterization of functional properties" deals with the isolation under optimal conditions of proteins obtained from scorpion bodies. The subject is sound and in general terms the manuscript is well structured and most results are relevant and deserve publication in the Molecules journal.

Nevertheless some of minor points should be considered to improve this manuscript. These are listed below.

Grammar and spelling should be revised all through the text.

Answer:

We tried to improve grammar and spellings all through the text.

Dried scorpions were grinded (line 65). Which was the granulometry of samples before extraction?

Answer:

Yes, when we grinded 10 g dried scorpions obtained powder passed through 40 mesh separator. We have include it in to the text.

Line 103. Experiments were run in duplicate, why not in triplicate as it is usual in experimental design and statistical analysis?

Answer:

Yes, we did this experiment in triplicate as usual. It was our technical mistake. We changed it in the text.

Please clarify In general terms there is some lack of discussion in the correspondinig section. I would encourage authors to compare their results with those obtained by other authors.

Answer:

Responding to your commitments we are agree with you and we fully changed context of manuscript, paying more attention to use in medicine. So we have added aim of work, why we are doing this work? and at which are might be useful obtained results? We explained in conclusions. Since most of your commitments were similar to the commitments of reviewer 1, we hope that, we have answered to your suggestions. If you read this manuscript again from the starting point, we hope that you will find responds to your commitments.

Also we did some changes to the title, authors and abstract.

Many tanks again for your support

Round 2

Reviewer 1 Report

Since the author have followed all comments which were given for the reviewers and the manuscript is finally improved, I can recommend the publication of this work.

Author Response

Dear Professor:

We would like to thank you for your kind comments and thank you very much again for you giving me a chance to publish my article.

Yours sincerely,

Corresponding author: Prof. Abulimiti Yili

Reviewer 2 Report

Dear authors,

It is not helpful that you do not provide me with a detailed description of the revisions I suggested. I cannot see your answers to reviewer 1, so this is not helpful at all to tell me we had similar remarks.

Author Response

Dear professor:

Thank you for your kind comment on our manuscript. We carefully read your reports and revised the manuscript according to the suggestions.

Response to Reviewer 2:

Your commitments:

The manuscript by Wali et al. describes the extraction of protein from dried scorpion tissue by ultrasonic treatment. While I believe the technical aspects of this paper are fine I have some concerns regarding the justification behind this technical development.

Specifically, the authors make claims in the abstract, introduction and discussion (e.g. lines 12-13; lines 39-42; lines 355-357) regarding the medical importance of consuming scorpion proteins, but they provide very little explanation to this claim.

Respond:

We clarified physiologically role of SP in the abstract, introduction and discussion sections. We added necessary references.

They provide some references, but still they should provide more details regarding the basis for their claims.

Respond:

We added new references about description of the biological role or activities of whole SP, unfortunately they are in Chinese language, but it is possible to read it by translation through Google Translate.

I do not mean any disrespect for traditional medicine, but this is a sensitive topic and scientific work should strive to make the most accurate and fact-based claims.

Respond:

Yes, you are right. We tried to do so.

Thus, I think the authors should significantly expand the description of the supporting evidence for their claims or alternatively tone them down.

Respond:

We tried to improve medicinal part, i.e., applications of SP in TCM and all the words, regarding to use it in food aspect removed.

Moreover, to the best of my knowledge much of the usage of scorpion in traditional medicine is specific to venom, but the authors in the current work do not focus on venom at all, but extract proteins from whole scorpion tissue.

Respond:

We added to Introduction section a sentence about our main goal was to focus to scorpion body’s whole protein extraction, we did not pay attention to separate venom.

Yours sincerely,

Corresponding author: Prof. Abulimiti Yili